# Evaluating Yield, Nutritional Quality, and Environmental Impact of Quinoa Straws across Mediterranean Water Environments

**DOI:** 10.3390/plants13060751

**Published:** 2024-03-07

**Authors:** Javier Matías, Verónica Cruz, María José Rodríguez, Patricia Calvo, Isaac Maestro-Gaitán, María Reguera

**Affiliations:** 1Agrarian Research Institute “La Orden-Valdesequera” of Extremadura (CICYTEX), 06187 Badajoz, Spain; veronica.cruz@juntaex.es; 2Technological Institute of Food and Agriculture of Extremadura (CICYTEX), 06007 Badajoz, Spain; mariajose.rodriguezg@juntaex.es (M.J.R.); patricia.calvo@juntaex.es (P.C.); 3Department of Biology, Campus de Cantoblanco, Universidad Autónoma de Madrid, c/Darwin 2, 28049 Madrid, Spain; isaac.maestro@uam.es

**Keywords:** climate change, quinoa valorisation, water environment, rainfed agriculture, livestock, byproducts

## Abstract

Quinoa (*Chenopodium quinoa* Willd.) is a promising and versatile crop due to its remarkable adaptability to diverse environments and the exceptional nutritional value of its seeds. Nevertheless, despite the recent extensive research on quinoa seeds, the straw associated with this crop has received comparatively little attention. The valorisation of this by-product provides an opportunity to improve the overall outcomes of quinoa cultivation. In this work, three quinoa varieties were evaluated for two years (2019 and 2020) under three different Mediterranean water environments (irrigation, fresh rainfed, and hard rainfed), aiming to assess the straw yield and nutritional quality and to study the changes in the crop nutritional uptake associated with different water environmental conditions. The nutritional analysis included the quantification of the ash, crude protein, crude fat, minerals (P, K, Ca, Mg), and fibre (gross fibre (GF), acid detergent fibre (ADF), neutral detergent fibre (NDF), acid detergent lignin (ADL), hemicellulose, cellulose) contents. As the results reveal, most of the parameters evaluated were susceptible to change mainly with the water environment but also with the genotype (or their interaction), including the yield, crude protein, relative feed value (RFV), and mineral content, which generally decreased under water-limiting conditions. Moreover, a comparative analysis revealed that straw Ca, Mg, and K contents were generally higher than in seeds. Overall, this study demonstrates that quinoa straw quality is genotypic and environmentally dependent, and these factors should be considered when aiming at improving straw feed value for livestock nutrition.

## 1. Introduction

In the 21st century, the world is confronted with a demographic challenge arising from continuous growth of the population, which is expected to reach 10 billion by 2050, putting food security at risk [1,2]. Furthermore, climate change poses a substantial threat to agricultural production. Projected future climatic conditions indicate a notable rise in temperatures and erratic rainfall patterns, causing a reduction in crop and herbage yields [3]. The shifting global climate patterns are leading to a scarcity of freshwater, which is particularly impactful on Mediterranean ecosystems. It is predicted that climate change will have a substantial impact on the Mediterranean region [4,5]. The associated effects are therefore projected to have repercussions on food security, emphasizing the pressing need for immediate measures to adjust agricultural practices to these evolving water environmental conditions [6,7]. Consequently, there will be a growing demand for nutrient-rich plant-based food and feed products derived from resilient crop varieties capable of withstanding challenging climatic conditions [8]. Moreover, there is significant competition for resources between livestock feeds and food production. However, by increasing the utilization of by-products and residues from the food system as animal feed, it is possible to mitigate this competition [9].

Quinoa (*Chenopodium quinoa* Willd.) has emerged as a promising and versatile crop with increasing global popularity, attributed to its exceptional adaptability to diverse environments and the outstanding nutritional value of its seeds [10]. Quinoa can grow under unfavourable soil and climatic conditions due to its tolerance to abiotic stresses, such as drought or salinity [11,12,13], and its huge genetic diversity [14,15,16]. Quinoa is, therefore, a well-recognized climate-resilient plant that constitutes an interesting alternative to traditional crops for new climate change scenarios [17,18,19,20,21,22,23,24]. In line with this, quinoa cultivation has witnessed a significant surge in recent years, accompanied by its spread to diverse regions throughout the world [25]. However, despite extensive research on quinoa seeds, the straw produced by this crop has received relatively limited attention. Hence, the valorisation of this by-product presents an opportunity to enhance and optimize quinoa cultivation. Indeed, the valuable resource of quinoa straw lies in its rich composition of lignocellulosic fibres, protein, and minerals, making it suitable for various applications, including use as animal feed [26]. 

To shed light on the untapped potential of quinoa straw, this study enhances our comprehensive understanding of how the yield and composition of quinoa straw can vary in response to different water environmental conditions and varieties. Additionally, this study aims to investigate the influence of rainfed cultivation on the nutrient uptake dynamics of quinoa in Southwest Europe. By examining these aspects, we seek to provide valuable insights into the sustainable production and utilization of quinoa in water-limited environments, contributing to the enhancement of agricultural practices and sustainability in Mediterranean environments.

## 2. Results

### 2.1. Straw Yield and Harvest Index (HI)

All factors (Y, WEC, and V) impacted significantly on both straw yield (kg ha^−1^) and harvest index (HI), as indicated in Table 1. In 2019, the average straw yield was 2076 kg ha^−1^ and the HI was 0.39, while in 2020, the average straw yield and the HI increased significantly to 2428 kg ha^−1^ and 0.42, respectively. Concerning the influence of WEC, the highest straw yield (2808 kg ha^−1^) was achieved under I. However, FR and HR resulted in lower straw yields (2081 kg ha^−1^ and 1868 kg ha^−1^, respectively), without significant differences between these two rainfed conditions. However, when considering the HI, similar values were obtained under I (0.43) and FR (0.44), with these being significantly higher both compared to HR (0.35). Among the varieties, Pasto and Marisma achieved the highest straw yield (2532 kg ha^−1^ and 2338 kg ha^−1^, respectively), while Titicaca exhibited the lowest value (1886 kg ha^−1^). On the contrary, the highest HI was obtained in Titicaca (0.45), this being notably greater than the HI of Pasto (0.38) and Marisma (0.39), which showed comparable values.

As shown in Appendix A, when examining the influence of the interactions among factors on straw yield and harvest index (HI), it was observed that in 2019, the lowest straw yield occurred with the FR treatment (1188 kg ha^−1^). Additionally, in 2019, there were notable differences among the varieties, with the straw yield of the Titicaca (1405 kg ha^−1^) being inferior to that found in Pasto (2468 kg ha^−1^) or Marisma (2214 kg ha^−1^). When analysing the interaction between WEC and V (WEC × V), WEC showed an impact on straw yield exclusively in Pasto. Indeed, significant changes were observed in Pasto between I (3096 kg ha^−1^) and HR (2144 kg ha^−1^) conditions. Importantly, it should be highlighted that the HI remained unaffected by WEC in the year 2020, and neither the Y × V nor WEC × V interactions influenced this factor.

### 2.2. Straw Composition

The nutritional composition of quinoa straws is presented in Table 2 and Appendix A. Significant differences were observed among the various factors analysed for ash (%), protein (%), fat (%), phosphorus (P %), potassium (K %), calcium (Ca %), and magnesium (Mg %) contents; gross fibre (GF %); neutral detergent fibre (NDF %); acid detergent fibre (ADF %); acid detergent lignin (ADL %); hemicellulose (%); cellulose (%); and relative feed value (RFV). 

When comparing between years (2019 and 2020), significant differences were observed in CF, K, Mg, GF, FND, ADF, ADL, cellulose, and RFV (Table 2). The CF content was lowered to half in 2020 (from 2.0% in 2019 to 1% in 2020). Also, the two minerals that yielded differences between years, Mg and K, were found in lower amounts in 2020, with a decrease to 0.68% in 2020 from 0.97% in 2019 in the case of Mg, and a decrease to 5.43% from 6.08% for K. In contrast, in 2020, the contents of GF, FND, ADF, and cellulose (31.4%, 49.1%, 35.8%, and 22.4%, respectively) were higher than in 2019 (22.7%, 36.8%, 25.3%, and 11.8%, respectively). It should be noted that the RFV in 2019 (183.5) was notably higher than in 2020 (116.6). When analysing the impact of the WEC on the straw composition, significant variations were observed in almost all parameters studied except for CF, Ca, and hemicellulose. The HR condition exhibited the highest ash content (19.0%), while the I and FR conditions showed lower values (14.1% and 14.3%, respectively). The crude protein (CP) content was noticeably lower under HR (9.3%) than under I (15.5%) and FR (14.3%) conditions. Both P and Mg showed a similar trend to the CP, displaying significantly higher contents under I (0.31% and 1.03%, respectively) and FR (0.25% and 0.94%, respectively) compared to HR (0.12% and 0.43%, respectively). The RFV was also significantly lower under HR (123.8) than under I (155.9) and FR (170.4), which showed statistically similar values to each other. In contrast, the highest levels of K, GF, FND, and ADF (6.25%, 29.7%, 47.5%, and 34.2%, respectively) were found under HR. In the case of K, similar levels were achieved under HR and FR. Regarding the impact of the variety (V) on the straw composition, significant differences were observed in the content of ash, CP, CF, K, Ca, and Mg. It is worth noting that the fibre composition (GF, FND, ADF, ADL, Hem., and Cell.) was not influenced by the variety. Pasto had the lowest ash content (14.5%), while Titicaca (16.5%) and Marisma (16.4%) achieved the highest levels. The lowest CP content (11.2%) was found in Pasto, whereas Titicaca (14.6%) and Marisma (13.4%) exhibited the highest values, which were similar between them. The CF content was slightly higher in Pasto (1.6%) than in Titicaca (1.4%), while Marisma achieved intermediate values (1.5%). The K content in Titicaca (6.05%) was higher than in Pasto (5.56%), while Marisma did not show differences in its K content when comparing among varieties (5.67%). The Ca content was higher in Pasto (2.28%) compared to Titicaca (2.06%), and, again, Marisma did not show significant differences compared to Titicaca or Pasto (2.33%). However, the Mg content in Marisma (0.90%) was higher than in Pasto (0.76%) and Titicaca (0.75%).

When examining the influence of the interactions among factors on straw composition, the interaction between year (Y) and water environmental conditions (WEC) (Y × WEC) revealed that in 2020, the P, FND, and Cell. contents were not affected by the WECs (Appendix A). However, in 2019, notable differences were identified among the three WECs. The highest P content was observed under I conditions (0.36%), whereas the lowest P content was recorded under HR (0.07%). In the case of FND and Cell., the highest levels were found under HR, reaching 43.8% and 26.0%, respectively. Conversely, the values found under I (34.2% for FND and 18.8% for Cell.) and FR (32.3% for FND and 17.7% for Cell.) were similar between them. The year (Y) and variety (V) interaction (Y × V) revealed that the P content displayed a decrease in Titicaca (0.16%), compared to Pasto (0.25%) or Marisma (0.29%), which exhibited comparable P levels. However, K and Mg did not yield variations among varieties within each year. The interaction between WEC and V (WEC × V) affected the contents of ash, CP, ADF, and Cell. On the contrary, this interaction unveiled no differences among varieties for these parameters under HR conditions. Also, the CF, ADL, and Hem. were not influenced by this interaction (WEC × V).

### 2.3. Relative Feed Value (RFV)

The RFV showed changes depending on the cultivation year (Y), the WEC, and the V (Table 2). The highest RFV was found in Marisma (155.9) and Titicaca (153.5), with significant differences observed between Marisma and Pasto (140.8). Between the years, the RFV was higher in 2019 (183.5) than in 2020 (116.6), and it achieved a lower value under HR (123.8) than under I (155.9) or FR (170.4) conditions. 

Also, the Y × WEC interaction influenced this parameter (Appendix A). Although the RFV remained unaffected by the WEC in 2020, the trend was lower RFV values under HR conditions than those that occurred in 2019. 

### 2.4. Nutrient Uptake, Nutrient Utilization Efficiency, and Nutrient Harvest Index

Nutrient uptake (U) was generally affected by the three factors analysed (the Y, the V, and the WEC) and, in most cases, by their interactions (Table 3). The Y had a significant influence on the NU, PU, and KU, with these being higher in 2020 (100.2, 13.6, 153.5 kg ha^−1^, respectively) than in 2019 (73.0, 8.5, 133.4 kg ha^−1^, respectively). The NU for all the mineral elements studied (N, P, K, Ca, and Mg) were higher under I conditions (115.8, 15.8, 159.1, 57.7 kg ha^−1^, respectively). In the case of N, P, and Mg, the NU achieved the lowest levels under HR (56.8, 5.6, and 4.6 kg ha^−1^, respectively). However, the NU of K and Ca were similar under HR (131.8, 45.4 kg ha^−1^, respectively) and FR (139.5, 46.2 kg ha^−1^, respectively) conditions. The V showed a significant influence on the NU of P, K, Ca, and Mg, with these being significantly lower in Titicaca than in Pasto or Marisma. The NU of N was not affected by the variety. 

The nutrient utilization efficiency (UtE) was affected by the factors analysed here in particular cases. Thus, the year presented a significant impact on CaUtE, with higher values in 2020 (36.0%) than in 2019 (28.1%). The WEC affected significantly PUtE, KUtE, CaUtE, and MgUtE, but the NUtE remained stable. The PUtE and MgUtE were significantly higher under HR (215.2 and 107.8 kg·kg^−1^, respectively) conditions than under I (131.5 and 36.6 kg·kg^−1^, respectively) or FR (157.7 and 75.0 kg·kg^−1^, respectively) conditions. On the contrary, the KUtE and CaUtE were lower under HR (8.2 and 23.5 kg·kg^−1^, respectively) conditions than under I (13.1 and 36.6 kg·kg^−1^, respectively) or FR conditions (11.8 and 36.1 kg·kg^−1^, respectively). 

The nutrient utilization efficiency (UtE) was also affected by the factors’ interactions, providing insights into the intricate relationships between them. The Y × WEC interaction influenced the PUtE and KUtE, showing that the WEC affected these two parameters only in 2019, resulting in significantly higher PUtE values under HR conditions (264.8 kg·kg^−1^) than under I (124.8 kg·kg^−1^) or FR conditions (191.8 kg·kg^−1^). The opposite pattern was observed for the KUtE, which showed lower values under HR conditions (5.7 kg·kg^−1^) than in I (11.8 kg·kg^−1^) or FR conditions (11.9 kg·kg^−1^). The Y × V interaction did not impact the nutrient UtE of the studied mineral elements, and the WEC × V interaction only had a significant effect on the MgNUtE, revealing that the value for Titicaca under HR conditions (148.9 kg·kg^−1^) was considerably higher, generally doubling the values when compared with the other varieties’ water environments (Appendix A). 

The results regarding the nutrient harvest index (HI) indicated significant effects on NHI, KHI, and MgHI due to the Y factor. The NHI was significantly higher in 2020 (0.49) compared to 2019 (0.43). The WEC impacted the KHI and CaHI, with these being, in both cases, lower under HR conditions (0.09 and 0.03, respectively) than under I or FR conditions (0.14 or 0.04, respectively). The nutrient HI was not influenced by the variety. Interactions among the factors were found to impact the nutrient HI only in the Y × WEC interaction for both NHI and KHI. In 2020, NHI experienced a significant increase under HR conditions (0.57) in contrast to I (0.46) or FR (0.44) conditions. Conversely, no significant variations in the NHI among the WECs were observed in 2019. As for KHI, in 2019, KHI exhibited lower values under HR conditions (0.06) compared to I (0.11) or FR (0.13) conditions, while in 2020, KHI remained consistent across all WECs.

### 2.5. Correlations and Principal Component Analysis (PCA)

A Pearson correlation coefficient analysis was performed to elucidate significant correlations between the parameters measured in quinoa straws per year (*p* < 0.05) (Figure 1). Then, a comparative analysis was considered, and clusters of positive and negative correlations that were maintained between years were identified. These clusters included strong negative correlations between RFV and GF, NDF, ADF, ADL, hemicellulose, and cellulose (with r = −0.794, −0.968, −0.955, and −0.502, respectively, in the 2019 samples, and with r = −0.867, −0.99, −0.978, and −0.578, respectively, for correlations in the 2020 samples) (Figure 1). Also, negative correlations were found between Mg and GF, NDF, ADF, and Cell. (with r = −0.788, −0.883, −0.845, and −0.881, respectively, for 2019, and r = −0.604, −0.513, −0.566 and −0.648, respectively, for 2020). On the other hand, positive correlations included strong correlations between straw yield and all nutrients’ uptake, between the fibre quality-related parameter pairs (GF, NDF, ADF, ADL, hemicellulose, and cellulose), between the different mineral uptakes’ parameter pairs, and between the different nutrient UtE and the minerals’ HI pairs (r values included in Appendix A). In the case of the cluster that included the relations between the HI with the nutrients’ UtE or with the minerals’ HI, 2020 showed positive correlations for all (r = 0.743 with NUtE, r = 0.444 with PUtE, r = 0.875 with KUtE, r = 0.814 with CaUtE, r = 0.508 with MgUtE, r = 0.625 with NHI, r = 0.632 with PHI, r = 0.868 with KHI, r = 0.618 with CaHI, and r = 0.656 with MgHI), but in 2019, only the positive correlations between the HI and the NUtE, KUtE, CaUtE, NHI, KHI, and CaHI were maintained (r = 0.963, 0.874, 0.686, 0.85, 0.581, respectively). Indeed, some correlations changed between the years, such as the negative correlation between the straw yield and certain minerals’ UtE or the straw yield with some minerals’ HI in 2020 (r = −0.492, −0.606, −0.802, −0.757, −0.507, and −0.627 with NUtE, PutE, MgUtE, NHI, PHI, and MgHI, respectively), correlations that did not appear in 2019. Also, in the case of Mg and hemicellulose, a significant correlation was only found in 2019 (r = −0.662), not in 2020. Furthermore, GF, NDF, ADF, and ADL yielded negative correlations with the mineral uptakes, specifically in 2020 (Appendix A). Noteworthy variations in the minerals’ uptake correlations included positive correlations in 2019 with ADL (r = 0.558 with NU, r = 0.591 with PU, r = 0.735 with KU, r = 0.56 with CaU, and r = 0.506 with MgU) and negative correlations in 2020 with Cell. content (r = −0.449 with NU, r = −0.502 with PU, r = −0.467 with KU, r = −0.548 with CaU, and r = −0.639 with MgU). 

The principal component analysis (PCA) reduced the dimensions of our data to three main components which were able to explain 80.73% of the variance (Figure 2). Component 1 contributed to 32.39% of the total variance. It was mainly explained, positively, by Cell., ADF, DNF, and GF (0.985, 0.978, 0.977, and 0.957, respectively) and by RFV, Mg, and CF, negatively (−0.956, −0.814, and −0.698, respectively). Component 2 contributed to 31.18% of the total variance, including variables with a positive impact, such as HI, KutE, and KHI (0.938, 0.96, and 0.9, respectively), and, negatively, Ash, PUtE, and K (−0.698, −0.628 and −0.535, respectively). Finally, component 3 contributed to 17.16% of the variance and was mainly explained by straw yield (0.96), KU (0.904), NU (0.853), NHI (−0.601), K (−0.579), and Ca (−0.573) (Appendix A). 

The reduction in dimensions and representation of the two principal components 1 and 2 in a biplot graph (Figure 2) revealed three main groups. Group 1 grouped all the 2020 samples, showing positive values for both component 1 and component 2, with high values of HI, GF, NDF, ADF, CaUtE, KHI, and MgHI (Table 1, Table 2 and Table 3). Group 2 contained all the 2019 samples under FR and I conditions, with negative values for component 1 and positive values for component 2. Thus, these samples showed high contents of Mg and RFV and excluded the 2019 HR samples, which showed lower values for these variables (Figure 2, Table 2). Group 3 included all HR samples, with negative values for component 2 and thus high values for ash and K contents, MgUtE, and PUtE and low values for KUtE and CaHI when compared to the FR and I samples (Table 2 and Table 3). Furthermore, within the Group 3, we could differentiate two subgroups, separating the 2019 and 2020 samples, with component 1’s values close to 0 in the case of the 2019 samples and positive values for component 1 in the case of the 2020 samples (Figure 2). In this group, the main variables separating the samples by year were GF, NDF, ADF, CaUtE, and KHI, which showed higher values in the 2020 samples when compared to the 2019 samples (Table 2 and Table 3). 

### 2.6. Mineral Content Comparison between Straws and Seeds 

When comparing the mineral content of straws and seeds (seeds obtained from the same experiment; data available in [27]), a log2FC analysis was applied. This analysis showed that all minerals evaluated except for P appeared consistently at higher concentrations in straws than in seeds (*p*.adj < 0.05) (Appendix A), with a log2FC > |1|, meaning the content values were at least doubled (Figure 3). This increase was particularly relevant for Ca content, as, in most cases, it reached a log2FC < (−2), indicating that the Ca content was 16 times higher in straws than in seeds (Figure 3). The Mg content was also significantly enriched in straws (generally 2–4 times higher) (*p*.adj < 0.05), except for Pasto harvested in FR conditions in 2019 (*p*.adj = 0.082) (Appendix A). 

The only mineral that was overall enriched in seeds compared to straws was P (*p*.adj < 0.05). However, no samples showed log2FC values exceeding |1|, indicating smaller differences between the straws and seeds in P content compared to the rest of the minerals (Figure 3). 

## 3. Discussion

Within the context of climate change, quinoa has been promoted as an emergent crop with the potential to contribute to food security worldwide, mainly attributed to its capacity to withstand abiotic stressors like salinity or drought [11,12,13] and the good nutritional quality of its seeds [28]. In line with this, currently, it is widely recognized that promoting greater environmental sustainability in agriculture requires the valorisation of agricultural waste and byproducts. Nevertheless, the potential of quinoa straw remains largely untapped, as the emphasis on quinoa cultivation has predominantly focused on the use of, and improvement in, seed yield. Thus, to further deepen the potential use of quinoa straws in agriculture, this study aimed to assess differences in straw production and composition, as well as nutrient uptake and utilization efficiency, among three quinoa varieties grown under different soil water conditions that included rainfed environments representative of Mediterranean agriculture. Moreover, a comparison between the mineral composition of seeds and straws was performed to assess changes in the mineral distribution due to genotypic or water environment differences. It is important to note that this study was conducted over two consecutive years, which presented notable differences in rainfall. The rainfall registered during the vegetative growth period (March to May) in 2020 (191.3 mm in the experimental station of La Orden, 151.2 mm in the experimental station of Maguilla) was approximately double that of 2019 (91.4 mm in La Orden, 76.4 mm in Maguilla). Additionally, the rainfall of Maguilla was around 20% lower than that of La Orden in both years, contributing to the harsher rainfed conditions of Maguilla (HR) as opposed to La Orden (FR). With this in mind, the main findings indicated that the straw performance and composition, as well as the crop nutrient utilization efficiency (UtE), were notably influenced by the water environmental conditions (WECs) and the quinoa variety (V) studied. Thus, the average straw yield was approximately 17% lower in 2019 (2076 kg ha^−1^) than in 2020 (2428 kg ha^−1^) as a consequence of the differences in the rainfall patterns between the years. This study gives slightly higher straw yields compared to the values achieved by the authors in a previous study performed in La Orden (1933 kg ha^−1^), in which six quinoa varieties were evaluated for two years under Mediterranean irrigated conditions [26]. It is well known that plant biomass decreases with increasing water limitation [29,30], and this would explain why the straw yield was significantly lower under rainfed (FR: 2081 kg ha^−1^; HR: 1868 kg ha^−1^) compared to irrigated conditions (I: 2808 kg ha^−1^). Indeed, generally, crop performance and productivity are significantly influenced by different environmental factors, with interannual variations observed locally [31]. However, when considering the Y × WEC interaction, it was noticed that in 2020, there were no straw yield differences between the I and FR conditions. This can be explained by the fact that the rainfall recorded during the cultivation period in La Orden (FR) was probably sufficient to prevent water stress, avoiding yield penalties. This finding confirms the tolerance of quinoa to water scarcity reported by others [12,20,32]. In line with this, the straw yield was similar in both years under HR conditions (2094 kg ha^−1^ in 2019; 1762 kg ha^−1^ in 2020). Furthermore, it was lower in 2019 under FR (1188 kg ha^−1^) than under HR (2094 kg ha^−1^) conditions. This could be attributed to the greater ability of the soil in Maguilla (HR) to retain rainwater during the winter months (the winter rainfall patterns were 168.4 mm in Maguilla in 2019 and 153.7 mm in 2020, and 187.4 mm in La Orden in 2019 and 208.1 mm in 2020) due to its higher clay content, in contrast to the sandy loam soil in La Orden. Therefore, the availability of soil water reserves probably played a role in mitigating water-related stress during the vegetative phase under HR conditions in 2019, minimizing the straw yield decrease, even though rainfall was insufficient in 2019 with an optimal fructification, as evidenced by the lower HI obtained (0.28 in 2019 and 0.41 in 2020).

Crop performance is a complex trait influenced by various factors in addition to the environment, such as the genotype and the interaction between them [33]. In this study, the variety also impacted the straw yield, with lower average yields found in Titicaca (1886 kg/ha) than in Pasto (2532 kg/ha) or Marisma (2338 kg/ha). However, when examining the Y × WEC interaction (Appendix A), it was observed that differences among varieties were significant only in 2019, which can be explained by the lower drought tolerance of the Danish-bred variety (Titicaca) than the Dutch-bred varieties (Pasto and Marisma), despite what was observed in a study conducted in [34]. 

The harvest index (HI) is a useful parameter that denotes the efficiency of partitioning biomass into harvested products, showing the balance between source and sink tissues [35,36]. In this study, the HI was higher in 2020 (0.42) compared to 2019 (0.39). Furthermore, the HI was significantly lower under HR (0.35) than under FR (0.44) and I (0.43) conditions. These disparities can be attributed to the influence of the water-limiting environment on quinoa growth and flowering, as observed by [37]. Regarding the influence of the variety on biomass partitioning, it was observed that Titicaca achieved a significantly larger HI (0.45) than Pasto (0.38) and Marisma (0.39). Considering a previous work that showed that high temperatures can affect quinoa fruit development, leading to a significant reduction in the HI [35], the higher HI of Titicaca could be related to the tolerance of this variety to heat stress, as also reported by [38]. 

Furthermore, it is crucial to investigate potential changes in the nutritional composition of the straw depending on the environmental conditions. Some studies have already indicated certain changes in the straw composition linked to the straw plant species and interannual variation [39,40]. However, when it comes to quinoa straw composition, very little research has been performed to date [26]. This work tries to shed light on the matter. The results show that the straw composition was influenced by both the year and the variety, in addition to the water environmental conditions studied. Indeed, there were significant differences between the years in CF, K, Mg, GF, NDF, ADF, ADL, Cell., and RFV. In 2020, lower levels of CF, K, Mg, and RFV were reached, which could be related to the higher straw yield achieved and the increased recorded precipitation. Thus, the higher CF content may result from a dilution effect due to the higher yield achieved. In 2020, fibre-related parameters (GF, NDF, ADF, ADL, and Cell.), except Hem., were higher compared to 2019. This could be attributed to the higher straw yield achieved in 2020, as was observed in a previous study [26], which probably required a greater presence of supporting tissue, such as fibre [41]. The higher content of NDF and ADL in 2020 caused lower dry matter intake (DMI) and digestible dry matter (DDM) values, respectively, resulting in a lower RFV compared to 2019. Noteworthily, in 2020, the average fat content (1%) was half that of 2019 (2%), which could be related to the higher yield achieved in that year and a dilution effect. The nutritional composition of quinoa straw exhibits variability based on crop management practices, as elucidated by Zulkadir et al. (2021) [42]. Their investigation revealed that the sowing date exerted a significant influence on the majority of the examined parameters, with the row spacing only impacting the P content. Comparable average P levels were identified between their study and ours. However, our study demonstrated a substantially higher average K content (5.74%) in contrast to the value (1.44%) reported by Zulkadir et al. (2021) [42]. Likewise, elevated average contents were observed for Ca (2.23% vs. 1.82%) and Mg (0.83% vs. 0.54%). These disparities in mineral content may be ascribed to variations in environmental conditions and genotypes.

Drought stress disrupts plant mineral nutrition [30], yet there is a lack of concrete knowledge about the exact effects of drought on the uptake of mineral nutrients and the consequent impacts on plant physiology [43]. In line with this, the greater K content in 2019 may be related to its role in drought resistance via acting as an osmoregulatory element [44], as the 2019 precipitation was considerably lower than that recorded in 2020. This is consistent with the results when comparing WECs, as the average K content was higher under FR (5.83%) and HR (6.25%) than under I (5.20%) conditions. Also, lower contents of P and Mg under HR could be attributed to a reduced demand for these elements under this water condition due to the reduced growth (straw yield). Indeed, Mg and P reductions have been observed in plants subjected to drought, although the exact mechanisms that result in this response remain largely unidentified [45,46]. It should be noted that P and Mg are required for protein synthesis in plants [47,48,49]. Intriguingly, the Mg content was correlated with the straw protein content in both years (r = 0.6, on average) and only P and protein in the driest year, 2019 (r = 0.7). 

One prospective application of quinoa straw involves its utilization as animal fodder [26]. Hence, considering its potential use in livestock feed, one of the paramount nutritional parameters is the protein content, which was influenced by the WECs, as similarly noted in quinoa seeds [27]. However, contrary to what was found in seeds, the straw protein content (CP) was significantly lower under HR (9.3%) than under I (15.5%) or FR (14.3%) conditions. Furthermore, the average protein content under I conditions was higher than that determined by the authors in a previous study under similar conditions (10.6%), although this previous study used different quinoa varieties [26]. Moreover, CP, the average contents of ash, P, Mg, GF, FND, ADF, and ADL were similar under I and FR conditions, but differed from HR conditions, probably due to the stronger water stress under this water environmental condition. Moreover, the greater levels of NDF and ADL under HR led to a lower RFV in the straw obtained in that case. Therefore, considering the lower CP content and RFV of the straw under HR conditions, the quality of the quinoa straw intended to be used for animal feed would worsen when obtained in this type of water environment. Furthermore, as can be observed in Table 2, the straw contents of CP, P, and Mg showed similar trends, achieving lower contents under HR conditions (9.3%, 0.12%, and 0.43%, respectively) than under I (15.5%, 0.31%, and 1.03%, respectively) or FR (14.3%, 0.25%, and 0.94%, respectively) conditions. In fact, the CP, P, and Mg contents were positively correlated with each other in straws (Figure 1, Appendix A), relations that were not observed in seeds [27].

Regarding the nutrient uptake of the main macronutrients (N, P, K), increased values were observed in 2020 (Table 3). These results can be partially explained by the higher seed and straw yield found in that year ([27] and this work) and the abovementioned mineral content differences. Thus, significantly higher levels of P and K were reached in seeds in 2020 (46.4% and 13.5%, respectively) compared to 2019, while higher levels of K and Mg were present in straws harvested in 2019 (11.9%, 42.6%, respectively) compared to 2020. Nutrient uptake (NU), except for N, was significantly higher in Pasto and Marisma than in Titicaca, which could be due to the higher straw yield achieved by these varieties (Table 1), as seed yield was similar [27]. In terms of nutrient utilization efficiency (UtE), the results show that only Ca was influenced by the year, with higher values in 2020. This could be attributed to the increased precipitation in that particular year since, as is discussed later, CaUtE was found to be higher under I conditions compared to HR conditions. NUtE demonstrated remarkable stability, exhibiting no variations with any of the studied factors or their interactions. This implies that N consumption follows a linear trend with biomass production, at least within the yield intervals attained in this study, without being influenced by the WEC or the variety. However, for the other examined minerals (P, K, Ca, and Mg), the WEC significantly impacted their UtE. In the case of P and Mg, UtE was higher under HR conditions than under FR or I conditions, potentially indicating reduced absorption of P and Mg under soil water stress [50,51]. On the contrary, KUtE and CaUtE were higher under I and FR than in HR, which could be linked to the induced drought tolerance mechanisms that have been reported to involve K [44] and Ca [52,53]. These variations in UtE based on the WECs need to be considered in the fertilization regimen for quinoa cultivation. It should be noted that, under I conditions, a reduced input of K and Ca per unit weight of seed produced and a larger input of P and Mg would be necessary compared to arid dryland conditions. Moreover, an impact of variety on the UtE was observed for P and Mg in Titicaca, with higher values compared to the other varieties, which could be potentially attributable to the distinct origin of the variety, as previously elucidated.

Regarding the nutrient partitioning in the plant, it is noteworthy that the studied factors and their interactions exerted minimal influence (Table 3). Only K and Ca were affected by the WEC, with lower KHI and CaHI values under HR conditions. This phenomenon can be elucidated by the plant’s tendency to accumulate higher concentrations of these elements in the stems under water limitation, potentially as a mechanism to enhance drought tolerance, as previously discussed. Notably, this study stands out as one of the few investigations where mineral composition is examined in both seeds and straws using identical plants (Figure 3). This allowed us to perform a comparative analysis between tissues, considering WECs, interannual variations, and genotypes, revealing that most of the minerals analysed were more abundant in straws than in seeds ([27] and this study), independently of the environmental conditions, genotypes, or years, except for P. Considering the characteristic mineral composition found in wheat grains [54,55] and straws [56,57], similar trends were observed for Ca (in larger amounts in straws than in seeds) but not for P, K, or Mg, which were quite similar between tissues. However, all of these previous studies did not compare tissues from the same plants, making it intriguing to conduct such a type of analysis. Interestingly, according to the results shown in this study, quinoa straws appear to be enriched in Ca, K, and Mg compared to the abovementioned wheat studies. 

## 4. Materials and Methods

### 4.1. Experimental Design and Plant Material 

During the 2019–2020 period, a field experiment was conducted in the Extremadura region (Southwest Spain) to evaluate three quinoa varieties adapted to European conditions (Pasto and Marisma, provided by Algosur S.L., Lebrija, Spain; Titicaca, supplied by Quinoa Quality, Copenhagen, Denmark) cultivated under three different water environmental conditions (irrigated (I), fresh rainfed (FR), and hard rainfed (HR)). The irrigated and fresh rainfed conditions were studied at the experimental station “La Orden” which belongs to the Center for Scientific and Technological Research of Extremadura (CICYTEX, Badajoz, Spain), located in the Guadiana Basin (lat. 38°51′10″ N; long. 6°39′10″ W). To study the HR water environmental conditions, the field experiment was carried out in a typical rainfed farm located in Maguilla (lat. 38°23′29″ N; long. 5°42′28″ W). Monthly mean minimum and maximum temperature (Tmin and Tmax) data, as well as rainfall data, were obtained from weather stations at the respective experimental stations “La Orden” and “Maguilla” (Appendix A). The climate in both years (2019 and 2020) aligns with the typical values for temperature and precipitation historically observed at the experimental stations. Also, to assess the plant stress level, water balance was estimated using precipitation and evapotranspiration (ET) values (Appendix A) obtained from the meteorological station near the experimental site. The experimental design was a split–split plot with four replications, with years as the main plot. The water environmental condition was the sub-plot, while the variety was the sub-sub-plot. Plants were collected when they reached physiological maturity. The experimental plots consisted of four rows, each 10 m long and spaced 0.75 m apart. Sowing took place in mid-February, with a seeding rate of 6 kg ha^−1^, utilizing a mechanical plot drill. In the irrigation treatment, a drip irrigation system was used to provide water, maintaining the soil under non-limiting water conditions. The total water input for each WEC is represented in Appendix A. The employed drip irrigation system utilized a 16 mm pipe with integrated emitters at 30 cm intervals, featuring a flow rate of 2 L/h. The separation between pipes was maintained at 1.5 m. Irrigation frequency during the period from 1 May to 14 May was set at 2 days per week, while during peak demand months (mid-May to mid-June), it increased to 3 days per week. In the preceding months, irrigation occurred once a week, tailored to the specific needs of the plants. In 2019, the harvesting dates for the fresh rainfed (FR) and hard rainfed (HR) conditions were 11 June and 19 June, respectively, while for the irrigated (I) conditions, the harvest was conducted on 10 July. In 2020, the harvest for the hard rainfed (HR) conditions was on 23 July, and for the fresh rainfed (FR) and irrigated (I) conditions, it was at the beginning of August (8 August). Seeds were separated from straw by a stationary thresher (Wintersteiger LD 352, Ried, Austria). 

The experimental plots at “La Orden” were characterized by a sandy loam texture with a pH of 6.9, 0.38% organic matter, 0.045 dS m^−1^ electrical conductivity, 0.24% total N, 93.4 ppm of P, 57.9 ppm of K, 2364 ppm of Ca, and 252 ppm of Mg. On the other hand, the soil of “Maguilla” was clayey soil with a pH of 7.6, 0.91% organic matter, 0.098 dS m^−1^ electrical conductivity, 0.26% total N, 67.8 ppm of P, 404.9 ppm of K, 6086 ppm of Ca, and 371 ppm of Mg. A fertilization rate of 150 kg ha^−1^ of N, 100 kg ha^−1^ of P_2_O_5_, and 100 kg ha^−1^ of K_2_O was applied. Based on the soil mineral composition and the prescribed fertilization rates, the levels of macronutrients were found to be non-limiting for quinoa growth in all the studied cases. Weeds were controlled mechanically. No significant pests or diseases were observed.

### 4.2. Straw Nutritional Quality Parameters

Analysis of contents of crude protein (CP), crude fat (CF) ash, minerals (P, K, Ca, and Mg), and fibre composition (gross fibre (GF), acid detergent fibre (ADF), neutral detergent fibre (NDF), acid detergent lignin (ADL), hemicellulose (Hem.), cellulose (Cell.)) along with the calculation of the relative feed value (RFV) and harvest index (HI) were conducted following the methodology described in [26].

Nutrient uptake (NU), nutrient utilization efficiency (UtE), and nutrient harvest index (NHI) were calculated according to Matias et al. (2021) [26].

### 4.3. Statistical Analysis

Data were analysed using the Statistix 8.0 analytical software (https://statistix.informer.com/8.0/, accessed on 1 August 2023) following a three-way analysis of variance (ANOVA), including the year (Y), water environmental conditions (WEC), variety (V), and their interactions in the model. The year was treated as a fixed factor. When the F ratio was significant (*p* < 0.05), the post hoc Tukey’s test was performed and used to compare means. 

A principal component analysis (PCA) dimensional reduction was performed following extraction by sedimentation and varimax rotation methods. The straw yield and HI were included in this analysis together with different straw nutritional quality-related parameters (including ash, CP, CF, P, K, Ca, Mg, GF, NDF, ADF, ADL, Hem., Cell., and RFV) and nutrient uptake, utilization efficiency, and nutrient harvest index-related parameters (including NU, PU, KU, CaU, MgU, NutE, PUtE, KutE, CaUtE, MgUtE, NHI, PHI, KHI, CaHI, and MgHI). This analysis was performed using SPSS 17.0 software. 

Correlations among variables dependent on the year factor were evaluated using a Pearson correlation coefficient test at a significance level of *p* < 0.05 applying the cor.mtest function, using the package corrplot v.0.92 [58]. Fold change ratios were calculated through the fold.change function, using the countdata package v.1.3 [59], and log2FC was calculated using the function foldchange2logratio from gtools package v.3.9.4 [60]. Statistical significance at *p* < 0.05 between straw and seed mineral content for each condition was analysed through a t-Student comparison, using the function *t*-test, and the *p*-adjusted was calculated applying the Benjamini–Hochberg correction for every comparison made per mineral using the function *p*.adjust. Both functions were obtained from the stats package, base R. These analyses were performed using R version 4.2.2 software.

## 5. Conclusions

In summary, this study provides novel information for optimizing quinoa cultivation with a focus on the sustainable and efficient use of quinoa straw for animal feed. The primary findings underscore the multifaceted influence of water environmental conditions and the genotypic factor on various aspects related to straw yield, composition, and nutrient dynamics. It was noted that environments characterized by water limitation may compromise both straw productivity and quality, along with affecting the uptake, utilization efficiency, and nutrient partitioning of the crop. Notably, lower protein (CP), Mg, and P contents, together with a diminished RFV, yielded poorer straw nutritional quality under harsh rainfed conditions, which may impact their suitability. Consequently, both the environmental factor and the genotype should be considered key determinants of the nutritional value of quinoa straws. 

## Figures and Tables

**Figure 1 plants-13-00751-f001:**
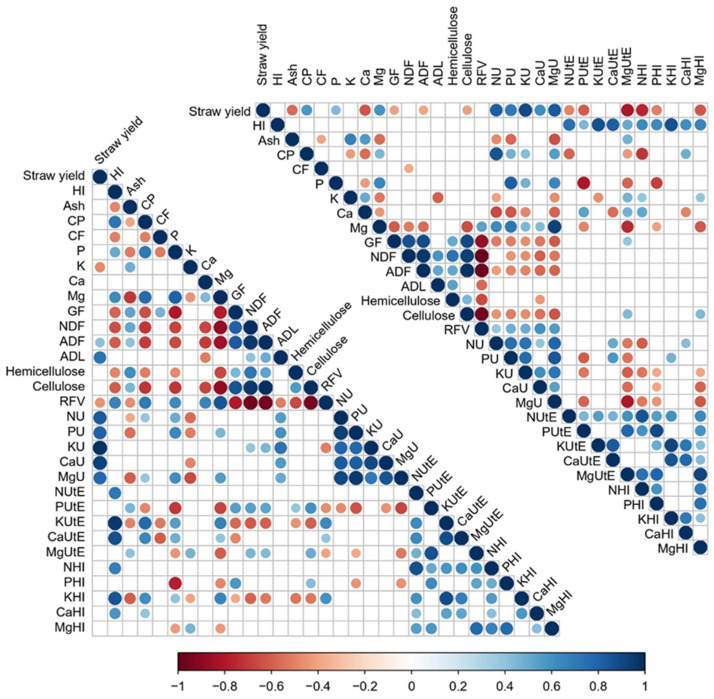
Correlogram of agronomical and nutritional parameters evaluated in quinoa straws. Correlogram of agronomical and nutritional parameters evaluated in quinoa straws in 2019 (**lower left** panel) and 2020 (**upper right** panel). The size of the circles represents, proportionally, the value of the Pearson correlation coefficient (r), and the colour represents the positive or negative values for this coefficient (blue and red, respectively). The parameters included in the correlogram correspond to yield-related parameters (straw yield and HI), nutritional composition-related parameters and relative feed value (Ash, CP, CF, P, K, Ca, Mg, GF, NDF, ADF, ADL, hemicellulose, cellulose, and RFV), nutrient uptake (NU, PU, KU, CaU, and MgU), nutrient use efficiency parameters (NUtE, PUtE, KUtE, CaUtE, and MgUtE), and minerals’ harvest index (NHI, PHI, KHI, CaHI, and MgHI).

**Figure 2 plants-13-00751-f002:**
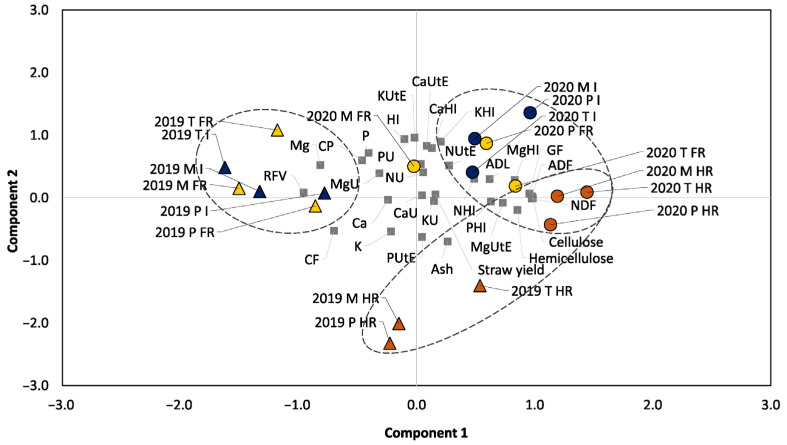
Principal component analysis (PCA) plotting components 1 and 2 for the different quinoa cultivars, environmental conditions, and years. Different colours represent the environmental conditions (with blue representing irrigated (I) conditions, yellow representing fresh rainfed (FR) conditions, and brown representing heavy rainfed (HR) conditions). Different shapes represent the year (with the triangles representing 2019 and the circles representing 2020). Component 1 (*X*-axis) explains 32.4% of the total variance. This component includes CP (−), CF (−), P (−), Mg (−), GF, NDF, ADF, ADL, hemicellulose, cellulose, RFV (−), MgUtE, NHI, PHI, and MgHI. Component 2 (*Y*-axis) explains 31.2% of the variance and includes HI, Ash (−), CP, CF (−), P, K (−), Mg, NU, PU, NUtE, PUtE (−), KUtE, CaUtE, KHI, and CaHI.

**Figure 3 plants-13-00751-f003:**
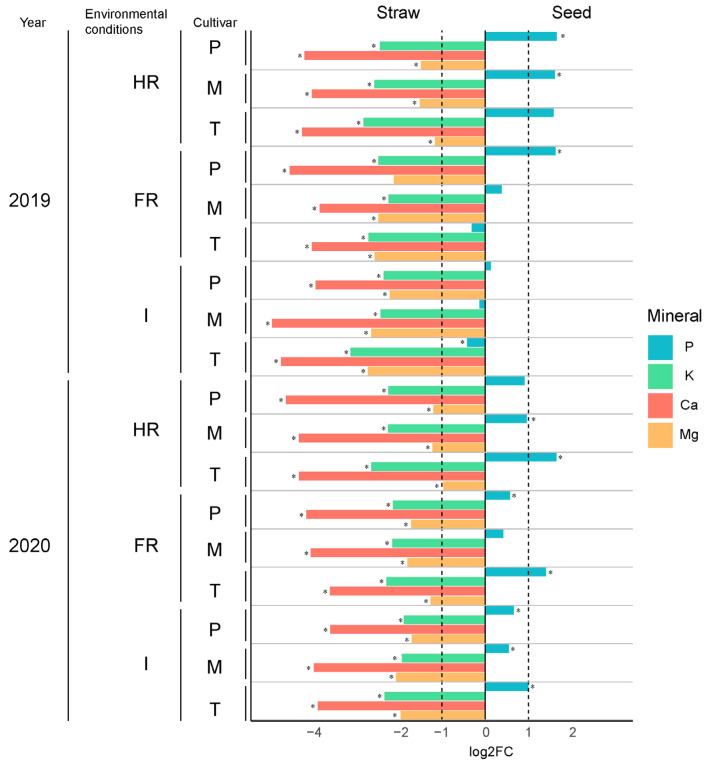
Comparison between quinoa straw and seed mineral composition. The Log2FoldChange (log2FC) was calculated for the comparison between straw and seed mineral contents for each year (Y), water environmental conditions (WEC), and variety (V). Different mineral contents are represented by different colours (with blue colour representing the P content, the green colour representing the K content, the pink colour representing the Ca content, and orange representing the Mg content). The asterisks (*) indicate significant differences in the mineral contents when comparing straw and seed contents after performing a t-Student test followed by BH correction (*p*-adjusted < 0.05).

**Table 1 plants-13-00751-t001:** Significance and means of straw yield (kg ha^−1^) and harvest index (HI) of three quinoa varieties (Pasto, Marisma, and Titicaca) grown under three water environmental conditions (I, FR, and HR) during two consecutive years (2019, 2020).

Treatment	Straw Yield (kg ha^−1^)	HI
Year (Y)	*	*
Water environmental conditions (WEC)	***	***
Variety (V)	***	*
Y × WEC	***	***
Y × V	***	n.s.
WEC × V	*	n.s.
Y × WEC × V	***	n.s.
Means		
Year (Y)		
2019	2076 b	0.39 b
2020	2428 a	0.42 a
HSD	302	0.02
Water environmental conditions (WEC)		
I	2808 a	0.43 a
FR	2081 b	0.44 a
HR	1868 b	0.35 b
HSD	386	0.03
Variety (V)		
Pasto	2532 a	0.38 b
Marisma	2338 a	0.39 b
Titicaca	1886 b	0.45 a
HSD	282	0.06

Different lowercase letters within the same column indicate significant differences at *p* < 0.05 according to Tukey’s test. HSD: critical value for comparison. n.s.: not significant; significant at * *p* < 0.05; and *** *p* < 0.001. I: irrigated. FR: fresh rainfed. HR: hard rainfed.

**Table 2 plants-13-00751-t002:** Nutritional composition and relative value of straw from three quinoa varieties (Pasto, Marisma, and Titicaca) grown under three water environmental conditions (I, FR, and HR) during two consecutive years (2019, 2020).

Treatment	Ash (%)	CP (%)	CF (%)	P (%)	K (%)	Ca (%)	Mg (%)	GF (%)	NDF (%)	ADF (%)	ADL (%)	Hem. (%)	Cell. (%)	RFV
Significance														
Year (Y)	n.s.	n.s.	**	n.s.	**	n.s.	*	**	**	**	*	*	**	**
Water environmental conditions (WEC)	***	***	n.s	n.s.	***	n.s.	***	**	**	***	*	*	***	***
Variety (V)	**	***	*	*	**	*	**	n.s.	n.s.	n.s.	*	n.s.	n.s.	*
Y × WEC	n.s.	*	n.s.	*	n.s.	***	*	n.s.	*	n.s.	n.s.	n.s.	*	**
Y × V	n.s.	n.s.	n.s.	**	*	n.s.	n.s.	n.s.	n.s.	n.s.	n.s.	n.s.	n.s.	n.s.
WEC × V	*	**	n.s.	n.s.	n.s.	n.s.	n.s.	n.s.	n.s.	*	n.s.	n.s.	*	n.s.
Y × WEC × V	n.s.	n.s.	n.s.	*	n.s.	n.s.	n.s.	n.s.	n.s.	n.s.	n.s.	n.s.	n.s.	n.s.
Means														
Year (Y)														
2019	16.0	13.1	2.0 a	0.23	6.08 a	2.32	0.97 a	22.7 b	36.8 b	25.3 b	4.4 b	12.5	11.8 b	183.5 a
2020	15.6	13.0	1.0 b	0.22	5.43 b	2.13	0.68 b	31.4 a	49.1 a	35.8 a	5.2 a	13.4	22.4 a	116.6 b
HSD	1.2	1.4	0.3	0.08	0.27	0.61	0.23	3.6	3.9	5.4	0.6	4.9	9.6	24.0
Water environmental conditions (WEC)														
I	14.1 b	15.5 a	1.3	0.31 a	5.20 b	2.08	1.03 a	25.6 b	41.7 b	29.5 b	5.1 a	12.2	17.3 ab	155.9 a
FR	14.3 b	14.3 a	1.6	0.25 a	5.83 a	2.27	0.94 a	25.7 b	39.6 b	27.9 b	4.4 b	13.1	13.4 b	170.4 a
HR	19.0 a	9.3 b	1.5	0.12 b	6.25 a	2.32	0.43 b	29.7 a	47.5 a	34.2 a	4.7 ab	13.5	20.6 a	123.8 b
HSD	1.9	1.6	0.4	0.10	0.04	0.39	0.21	2.9	3.8	2.9	0.6	3.1	4.7	15.9
Variety (V)														
P	14.5 b	11.2 b	1.6 a	0.22	5.56 b	2.28 a	0.76 b	27.8	44.3	31.7	5.0	12.6	19.1	140.8 b
M	16.4 a	13.4 a	1.5 ab	0.25	5.67 ab	2.33 ab	0.90 a	27.0	41.7	29.5	4.7	12.2	17.3	155.9 a
T	16.5 a	14.6 a	1.4 b	0.21	6.05 a	2.06 b	0.75 b	26.2	42.9	30.4	4.6	14.0	14.9	153.5 ab
HSD	1.5	1.5	0.2	0.05	0.38	0.23	0.11	1.8	2.6	2.7	0.4	3.1	5.2	14.3

Different lowercase letters within the same column indicate significant differences at *p* < 0.05 according to Tukey’s test. HSD: critical value for comparison. n.s.: not significant; significant at * *p* < 0.05; ** *p* < 0.01; and *** *p* < 0.001. I: irrigated. FR: fresh rainfed. HR: hard rainfed. CP: crude protein.

**Table 3 plants-13-00751-t003:** Nutrient uptake (U), use efficiency (UtE), and harvest index (HI) of three quinoa varieties (Pasto, Marisma, and Titicaca) grown under three water environmental conditions (I, FR, and HR) during two consecutive years (2019, 2020).

Treatment	NU (kg ha^−1^)	PU (kg ha^−1^)	KU (kg ha^−1^)	CaU (kg ha^−1^)	MgU (kg ha^−1^)	NUtE (kg·kg^−1^)	PUtE (kg·kg^−1^)	KUtE (kg·kg^−1^)	CaUtE (kg·kg^−1^)	MgUtE (kg·kg^−1^)	NHI	PHI	KHI	CaHI	MgHI
Significance															
Year (Y)	**	**	*	n.s.	n.s.	n.s.	n.s.	n.s.	*	n.s.	*	n.s.	**	n.s.	**
Water environmental conditions (WEC)	***	***	**	***	***	n.s.	**	***	***	**	n.s.	n.s.	**	**	n.s.
Variety (V)	n.s.	***	**	***	***	n.s.	**	n.s.	n.s.	**	n.s.	n.s.	n.s.	n.s.	n.s.
Y × WEC	***	***	***	***	***	n.s.	*	*	n.s.	n.s.	**	n.s.	*	n.s.	n.s.
Y × V	**	*	**	*	**	n.s.	n.s.	n.s.	n.s.	n.s.	n.s.	n.s.	n.s.	n.s.	n.s.
WEC × V	n.s.	*	n.s.	*	*	n.s.	n.s.	n.s.	n.s.	*	n.s.	n.s.	n.s.	n.s.	n.s.
Y × WEC × V	***	**	***	*	***	n.s.	n.s.	n.s.	n.s.	n.s.	n.s.	n.s.	n.s.	n.s.	n.s.
Means															
Year (Y)															
2019	73.0 b	8.5 b	133.4 b	47.0	21.1	17.5	193.8	9.9	28.1 b	71.2	0.43 b	0.50	0.10 b	0.03	0.13 b
2020	100.2 a	13.6 a	153.5 a	52.5	21.7	18.8	142.5	12.1	36.0 a	95.5	0.49 a	0.58	0.14 a	0.04	0.21 a
HSD	8.5	2.2	20.0	7.8	5.8	2.4	62.0	2.9	5.7	25.5	0.03	0.11	0.01	0.02	0.03
Water environmental conditions (WEC)															
I	115.8 a	15.8 a	159.1 a	57.7 a	32.1 a	17.6	131.5 b	13.1 a	36.6 a	67.3 b	0.44	0.48	0.14 a	0.04 a	0.15
FR	87.0 b	11.9 b	139.5 b	46.2 b	21.8 b	18.7	157.7 b	11.8 a	36.1 a	75.0 b	0.45	0.56	0.14 a	0.04 a	0.17
HR	56.8 c	5.6 c	131.8 b	45.4 b	10.5 c	18.1	215.2 a	8.2 b	23.5 b	107.8 a	0.49	0.58	0.09 b	0.03 b	0.19
HSD	13.1	2.4	18.2	9.4	4.6	2.8	42.3	1.6	4.9	21.7	0.06	0.14	0.02	0.01	0.04
Variety (V)															
P	85.5	12.4 a	154.6 a	55.5 a	23.3 a	18.7	155.5 b	10.7	29.7	76.5 b	0.47	0.54	0.12	0.04	0.17
M	92.1	12.4 a	150.1 a	55.2 a	24.8 a	18.1	152.0 b	10.7	29.5	71.8 b	0.45	0.51	0.13	0.04	0.15
T	82.1	8.5 b	125.7 b	38.6 b	16.2 b	17.6	197.0 a	11.7	36.9	101.8 a	0.46	0.57	0.11	0.04	0.19
HSD	12.1	2.1	20.0	8.5	3.5	3.0	33.3	2.9	10.9	19.6	0.07	0.08	0.03	0.01	0.04

Different lowercase letters within the same column indicate significant differences at *p* < 0.05 according to Tukey’s test. HSD: critical value for comparison. n.s.: not significant; significant at * *p* < 0.05; ** *p* < 0.01; and *** *p* < 0.001. I: irrigated. FR: fresh rainfed. HR: hard rainfed.

## Data Availability

Data are contained within the article or Appendix A.

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
