# Peer review of "Evaluating Yield, Nutritional Quality, and Environmental Impact of Quinoa Straws across Mediterranean Water Environments"

_plants, 2024, doi:10.3390/plants13060751_

Round 1

Reviewer 1 Report

Comments and Suggestions for Authors

The manuscript demonstrates a comprehensive and thorough investigation into the performance and composition of quinoa straw under different conditions and genotypes. This depth of analysis is a strong point, indicating the effort invested in the research. The exploration of the nutritional composition of quinoa straw adds novelty to the existing literature. The identification of potential agricultural uses for quinoa straw is a significant contribution that can benefit the scientific community and practitioners. 

The findings are presented in a clear and organized manner. The structure of the manuscript aids in understanding the complex interactions studied, making it accessible to a broad audience.

However, this article is very ordinary, not in the least fresh idea. Below are some suggestions for further improvements: 

-       Authors should check and provide evidence on whether 2019 and 2020 were 'typical' years weather-wise.? 

-       Clarify if pests, weeds, and diseases were controlled according to good practice. 

-       Provide details on soil water potential (soil water regime; in soil moisture (%) or FC%) and leaf water potential, and how soil was maintained during rainfall events in each condition throughout the quinoa life cycle?

-       There were multiple rainfall events during the drought stress period, how the soil was maintained for the appropriate ‘water limitation’, if no soil or leaf water status have been monitored and hence provided ?.

-       Have the authors considered performing a mean-centering of the correlation data before performing the PCA ?. Could the authors identify which traits linked to PC1 bi-directions (+ <> -) and PC2 bi-directions (+ <> -) ?

-       The authors should clarify drought resistance of the quinoa varieties and compare the traits between different groups. Otherwise, what was the objectives behind different water environmental conditions (I, FR, and HR) if the authors only focus on the nutritional quality (without taking into consideration the other factors variety and water status).

-       All detected traits did not present the correlation with the classification of drought resistance. We still don’t know which parameter(s) will be considered in improving (breeding or genetically-modified) the quinoa straw under water limitation. Therefore, the authors have not answered their purpose to provide details for functional traits to “optimize quinoa cultivation focusing on the sustainable and efficient use of quinoa straw for animal feed”

-       A more in-depth discussion on the studied factors and their impact on quinoa straw would strengthen the manuscript.

-       While the manuscript briefly mentions increased nutrient uptake in 2020, further exploration of patterns and implications is needed.

-       Provide additional details on the statistical analyses employed for comparing straw performance, composition, and nutrient uptake among different years, varieties, and water conditions.

Minor points:

-       Include recent literature on quinoa straw composition and utilization to provide a more comprehensive context for the study.

-       English should be improved for better quality and clarity.

-       The supp data should be provided as excel file (for easy-read/scroll) instead of PDF

Comments on the Quality of English Language

Extensive editing of English language required

Author Response

Dear Reviewer 1, first, we would like to express our gratitude for the revisions made to this manuscript. Your input has significantly contributed to the improvement of our work, and we truly appreciate the time and effort you have invested.

-       Authors should check and provide evidence on whether 2019 and 2020 were 'typical' years weather-wise.? 

Thanks for raising this point, Reviewer 1. As can be observed by comparing the climatograms of the years 2019 and 2020 for both locations, as depicted in the supplementary figure, with the corresponding climatograms representing the historical values of both locations shown in the figure below, it can be confirmed that the climate in both years (2019 and 2020) aligns with typical values for temperature and precipitations. However, the precipitations during the period from December 2018 to March 2019 were slightly lower than historical values in both locations. We have added this information on lines 534-536 of the Methods section.

PLEASE, SEE FIGURE IN THE ATTACHED PDF FILE

Climatogram of historical values for a) La Orden and b) Maguilla.

-       Clarify if pests, weeds, and diseases were controlled according to good practice. 

Thanks, Reviewer 1. Weeds were controlled mechanically. No significant pests and diseases were observed. We have added this information on Lines 559-560 (Methods section).

-       Provide details on soil water potential (soil water regime; in soil moisture (%) or FC%) and leaf water potential, and how soil was maintained during rainfall events in each condition throughout the quinoa life cycle?

Thanks, Reviewer 1. In this study, we did not use moisture probes or a Scholander chamber; however, this is an area of interest for our research, and in future studies, this valuable suggestion will be taken into consideration. Nonetheless, even without these data, we can assess the plant stress level by conducting a water balance using precipitation and evapotranspiration (ET) values (ET values included below) from the meteorological station near the experimental site. To do this, we consider the soil water retention values, which can be calculated using the following information: La Orden (Field capacity: 17.1% w/w; Wilting point: 7.9% w/w; Bulk density: 1.61 t/m³; Depth >1 m) and Maguilla (Field capacity: 37.0% w/w; Wilting point: 24.6% w/w; Bulk density: 1.45 t/m³; Depth >1 m).

Figure. Evapotranspiration (ET) in La Orden (a) and Maguilla (b) for the years 2019 and 2020.

The first paragraph of the discussion provides a rationale for this issue (See lines 355-362). Besides, we have included this information on Lines 536-588 of the Methods section and as a new Supplementary Figure (Supplementary Figure 2).

-       There were multiple rainfall events during the drought stress period, how the soil was maintained for the appropriate ‘water limitation’, if no soil or leaf water status have been monitored and hence provided ?.

Thanks, Reviewer 1. In the first year of the trial, precipitation was very limited in both locations during December 2018 and March 2019, as shown in the Supplementary Figure 1, especially in Maguilla, as explained in the first paragraph of the discussion. In the second year, precipitation increased, explaining why yields under the rainfed conditions were similar to those in the irrigated conditions. It is important to note that this work was not a water stress experiment but a study of cultivation under different representative Mediterranean water environmental conditions, and thus, we did not aim at controlling 'water limitation.' Not intervening in the soil water level is what allows us to speak about environmental conditions; otherwise, we would have to include specifically water regimes. This fact precisely highlights that under irrigated conditions, quinoa can achieve yields similar to those in the driest rainfed area, reflecting the low water requirements of quinoa. In future studies, a comprehensive examination of quinoa's response to different irrigation doses will be conducted, with monitoring of soil and leaf moisture levels as Reviewer 1 suggests.

The previous point serves to explain this question. Additionally, the stress state of the leaves could be visually observed based on leaf turgidity and the condition of the soil (visually and by touch).

-       Have the authors considered performing a mean-centering of the correlation data before performing the PCA ?. Could the authors identify which traits linked to PC1 bi-directions (+ <> -) and PC2 bi-directions (+ <> -) ?

Thanks, Reviewer 1. The mean-centering before conducting the PCA analysis was performed using Z-scores. This information was not included in the Methods section, as it is widely known and constitutes a standard procedure when performing this analysis. The traits linked to each Principal Component and their influence (both positive and negative loadings) are included in Supplementary Table 4 (Table S4. PCA rotated component matrix).

- The authors should clarify drought resistance of the quinoa varieties and compare the traits between different groups. Otherwise, what was the objectives behind different water environmental conditions (I, FR, and HR) if the authors only focus on the nutritional quality (without taking into consideration the other factors variety and water status).

Thanks, Reviewer 1. As we have mentioned previously, the main objective of the study was to assess the response of quinoa to different water environmental conditions (comparing an irrigation environment and two different rainfed conditions) in terms of straw yield and straw quality, aiming to revalue this crop recently established in Europe. Throughout the manuscript, we evaluated not only the environmental water condition (WEC) but also the genotypic factor (variety) and the interaction of both factors. Regarding the water stress resistance of the varieties studied in terms of seed yield penalties, it was assessed in a previous article conducted in the same experimental stations as those presented in this study (J. Matías et al., “Assessment of the changes in seed yield and nutritional quality of quinoa grown under rainfed Mediterranean environments,” Front Plant Sci, vol. 14, Nov. 2023, doi: 10.3389/FPLS.2023.1268014). On the other hand, the studied varieties were analyzed for their response to water stress in a greenhouse experiment by Maestro-Gaitán et al. 2023 ("Genotype-dependent responses to long-term water stress reveal different water-saving strategies in Chenopodium quinoa Willd.," Environ Exp Bot, vol. 201, p. 104976, Sep. 2022, doi: 10.1016/J.ENVEXPBOT.2022.104976). Both works have been cited in this manuscript and the yield penalty for these varieties, which was similar among them, was mentioned in Line 471, evidencing a similar water stress resistance among these varieties. Maestro-Gaitán et al. 2023 specifically evaluated the response to prolonged water stress of the three varieties analyzed here and assessed various physiological parameters that account for their stress level and tolerance (in terms of seed yield loss). Thus, in the present work, with a different scientific goal, both factors (WEC and genotype) have been taken into account, and the specific response to water stress is not evaluated as is not the objective of the research work. Instead, the focus is on the performance and straw quality of quinoa in three real agronomic environments, rainfed and irrigated conditions.

-       All detected traits did not present the correlation with the classification of drought resistance. We still don’t know which parameter(s) will be considered in improving (breeding or genetically-modified) the quinoa straw under water limitation. Therefore, the authors have not answered their purpose to provide details for functional traits to “optimize quinoa cultivation focusing on the sustainable and efficient use of quinoa straw for animal feed”

Dear Revisor 1, as we have mentioned previously, the objective of this work was not to assess the response to water stress but, by analyzing three varieties adapted to the Mediterranean field conditions, to evaluate the performance and nutritional quality of quinoa straw under different agronomic conditions (representative of the Mediterranean region: irrigation and rainfed conditions). Taking this into account, we have identified which parameters fluctuate with environmental conditions and, accordingly, determined what aspects to consider depending on the variety and environmental conditions. In general, we observe that under extreme rainfed conditions (HR), the quality of quinoa straw deteriorates. This is significant as it allows the establishment of a putative improvement range in parameters such as protein, which is crucial for animal nutrition. Furthermore, further research can focus on identifying ideal water conditions for specific quinoa varieties to maximize the quality of the resulting straw. Besides, considering the results of the PCA, samples are grouped according to the water environment (WEC), as the 'hard rainfed' (HR) clusters all the samples, while the rest are grouped based on the year (showing a different precipitation pattern between 2019 and 2020). For each group, each component exhibits a differential behavior, supporting the consideration of certain parameters for improvement depending on the water cultivation environment.

-       A more in-depth discussion on the studied factors and their impact on quinoa straw would strengthen the manuscript.

Thanks, Reviewer 1. We have carefully revised the Discussion section and we are unsure about which aspects should be discussed deeply. Nonetheless, we have discussed further some points that appear now marked in red throughout the text.

-       While the manuscript briefly mentions increased nutrient uptake in 2020, further exploration of patterns and implications is needed.

Thanks for this valuable suggestion, Reviewer 1. Following your recommendation, we have discussed further this important aspect. You will find new information on Lines 482-484 and on Lines 493-499.

-       Provide additional details on the statistical analyses employed for comparing straw performance, composition, and nutrient uptake among different years, varieties, and water conditions.

Dear Reviewer 1, details on the statistical analyses were included within the Methods section on lines 538-540 (subsection 4.1) and lines 571-575 (subsection 4.3). If any other information is missing, please let us know which points should be extended.

Minor points:

-       Include recent literature on quinoa straw composition and utilization to provide a more comprehensive context for the study.

Thanks, Reviewer 1. Sorry about this. We have revised recent literature on the topic as suggested and included the information in the Discussion section (Please, see lines 428-436): (Zulkadir, G., Ä°dikut, L., 2021. The impact of various sowing applications on the nutritional value of Quinoa Dry Herb. J. Food Process. Preserv. 45, e15730. https://doi.org/10.1111/JFPP.15730)

-       English should be improved for better quality and clarity.

Thank you, Reviewer 1. We have revised the text and made the necessary changes (highlighted in red throughout the text) to enhance the quality of the English as requested.

-       The supp data should be provided as excel file (for easy-read/scroll) instead of PDF

Dear Reviewer 1, we have uploaded the document as a PDF since the system did not allow us to submit an Excel file. If the article is accepted, we will upload the Excel file, as we acknowledge that the PDF may be challenging to read.

Reviewer 2 Report

Comments and Suggestions for Authors

The figures/tables/images are appropriate, they  are easy to interpret and understand, the data interpreted is appropriately and consistently throughout the manuscript, the results are processed the statistical analysis, the conclusions are consistent with the arguments presented

Author Response

Thank you very much, Reviewer 2. We greatly appreciate your assessment after evaluating our manuscript.

Reviewer 3 Report

Comments and Suggestions for Authors

I have reviewed the manuscript titled: Evaluating Yield, Nutritional Quality, and Environmental Impact of Quinoa Straws Across Mediterranean Water Environments

The manuscript is well written. I have only some minor suggestions:

- please edit the tables so they are easier to read, since now it is diffiuclut to follow

Author Response

First of all, we would like to thank your valuable comments, Reviewer 2. Regarding the edition of Tables, as we had to include the Tables within the template, we believe that the current format is not as clear as it should. We will make sure that they will appear improved in the final publishable version of the manuscript.

Round 2

Reviewer 1 Report

Comments and Suggestions for Authors

I have thoroughly reviewed the manuscript, and it appears that the authors have not adequately addressed the concerns and flaws raised during the initial review. While the reviewer identified some potential for the study to contribute to the research field, they highlighted critical pitfalls. Specifically, the reviewer identified methodological flaws related to the water regime (referred to as the water factor) and the measured/discussed parameters.

The results indicate that the majority of the evaluated parameters were sensitive to changes in the water regime, as mentioned by the authors in lines 24, 26, and throughout the text, as well as influenced by the genotype or their interaction.

The reviewer expressed concerns about the lack of a clear correlation between traits and the water regime, suggesting that different water environmental conditions (as seen in the comparison between an irrigation environment and two different rainfed conditions) were not adequately addressed. As it stands, additional data are required (basically soil water data or leaf potential), otherwise, a different focus with an entirely different design should be implemented, such as exploring different locations, soils, weather conditions, without solely concentrating on the water regime, are necessary.

Furthermore, the experimental design involving three quinoa varieties and three different water environmental conditions did not demonstrate a correlation with the classification of WEC. This raises questions about which parameter could be improved in breeding or targeted for GE. The authors have not sufficiently addressed their stated purpose of providing a general guide to functional traits and "genome" variation, particularly concerning WEC, as evident in their revised versions.

Comments on the Quality of English Language

English should be improved for better quality and clarity.

Author Response

Dear Reviewer 1,

We extend our sincere gratitude for your review of our manuscript. We appreciate the time and effort you invested in providing constructive feedback, which has undoubtedly contributed to the enhancement of our work.

We acknowledge your concern regarding the distinction between water stress response and the response to different water environmental conditions. It is our utmost priority to clarify this aspect further. Our study indeed delves into the diverse water environmental conditions and their impact on quinoa's adaptation and straw yield and quality. We aimed to offer a comprehensive understanding of how quinoa interacts with its surroundings, with a particular focus on the straw, an aspect that has been relatively underexplored in the existing literature.

Your insights regarding methodological flaws and the water factor have been duly noted. In this regard, we have made corrections within the Abstract, explicitly avoiding the term "water regime" and replacing it with "water environment" to eliminate any potential confusion. Furthermore, we want to emphasize that our study does not aim to determine the response to specific water stress conditions. We refrained from employing the term "water stress" without a probabilistic qualifier in the text, as we acknowledge the absence of direct measurements for the plant's water stress response. We appreciate your attention to detail and have taken measures to ensure clarity on this point throughout the manuscript. In line with this, in addressing your inquiry about soil water data or leaf potential, we regret to inform you that these specific measurements were beyond the scope of our study, as mentioned in our previous response to your review. Instead, we focused on key data such as precipitation regime, evapotranspiration, and irrigation (for irrigated conditions), which we believe are pivotal for comprehending the differences among the water environmental conditions analyzed. We have additionally incorporated soil characteristics, including texture, water retention capacity, and depth, providing comprehensive data to affirm the presence or absence of water scarcity periods. It is important to note that our analysis extends beyond the study of water regimes, encompassing broader implications. Hence, we have explored various environmental conditions typical of the Mediterranean to assess the potential expansion of quinoa cultivation. As evident from our findings, in fresh rainfed conditions (FR), different years yield varying results characterized by contingent on precipitation patterns. Our objective is not to analyze how quinoa responds to soil water content specifically but rather to investigate how quinoa adapts to diverse environments. Importantly, our focus is on understanding its performance in different water environments, where certain years may present more or less water-limiting conditions.

Concerning the comment on the experimental design and the correlation with the classification of Water Environmental Conditions (WEC), we may have misunderstood your point. Our statistical analyses aimed to identify parameters that change with water environmental conditions, the genotype (variety), or their interaction. The results (supported by the data and statistical analyses performed) indicate that numerous parameters, including yield, crude protein, relative feed value (RFV), and mineral content, were predominantly influenced by the water environment, with some discernible effects attributed to genotype or their interaction. Notably, these parameters exhibited a general decrease under water-limiting environments. Thus, the term "water environmental environment" in our manuscript is employed to delineate various locations characterized by distinct water availability. This designation aims to encapsulate the diverse settings with varying water conditions, specifically referring to locations with disparate water availability profiles.

Reviewer 1 has once again emphasized the necessity for English language revision. We are uncertain about the specific areas that may require improvement or clarification. It is important to note that the manuscript underwent a thorough review by a native English speaker to ensure linguistic accuracy. We would appreciate any specific guidance or examples that could assist us in addressing the language-related concerns raised by the reviewer.

Thank you once again for your valuable contributions to our manuscript,

Yours sincerely,

Maria
